# Caliche and Seawater, Sources of Nitrate and Chloride Ions to Chalcopyrite Leaching in Acid Media

**Pía Hernández** [1,*], **Giovanni Gahona** [1], **Monserrat Martínez** [1], **Norman Toro** [2,3] **and Jonathan Castillo** [4]

[1] Departamento de Ingeniería Química y Procesos de Minerales, Universidad de Antofagasta, Av. Angamos 601, Antofagasta 1270300, Chile; giox.190@gmail.com (G.G.); monserrat.martinez.v@gmail.com (M.M.)

[2] Faculty of Engineering and Architecture, Universidad Arturo Prat, Almirante Juan José Latorre 2901, Antofagasta 1244260, Chile; ntoro@ucn.cl

[3] Departamento de Ingeniería en Metalurgia y Minas, Universidad Católica del Norte, Av. Angamos 0610, Antofagasta 1270709, Chile

[4] Departamento de Ingeniería en Metalurgia, Universidad de Atacama, Av. Copayapu 485, Copiapó 1531772, Chile; jonathan.castillo@uda.cl

\* Correspondence: pia.hernandez@uantof.cl; Tel.: +56-55-2-637525

**Abstract:** Hydrometallurgical processing of chalcopyrite is of great interest today due to the depletion of oxidized copper minerals. This will also enable existing plants to continue operation. The objective of this work is to study the behavior of chalcopyrite leaching by stirring in an acid-nitrate-chloride media where seawater and brines provide chloride ions and nitrate ions can be provided from the caliche industry. The variables studied were sulfuric acid, nitrate and chloride concentration, source of water (dissolvent), temperature, solid/liquid ratio, particle size, mineral sample, and pretreatment before the leaching process. Despite being a refractory mineral, chalcopyrite can be leached in this system obtaining favorable recoveries at the conditions studied. It was possible to obtain 50% Cu in 0.7 M of $H_2SO_4$ and $NaNO_3$, using brine at 45 °C. The nitrate-chloride-acid system was highly temperature dependent, with an activation energy of 82.6 kJ/mol, indicative of chemical reaction control of leaching kinetics. SEM/EDS indicated the presence of sulfur on the surface of the mineral after leaching. This study demonstrates that sources such as seawater or discard brines (such as from the reverse osmosis process) and waste (solid or solutions) from the caliche industry can provide a highly oxidative system for the dissolution of chalcopyrite.

**Keywords:** chalcopyrite; leaching; seawater; chloride; nitrate

## 1. Introduction

Northern Chile is characterized by being a mining zone where the main industries are copper followed by industrial salts containing nitrates, iodine, lithium, among others [1]. Copper mining is made up of hydrometallurgical processes for oxidized minerals and pyrometallurgical processes for sulfide minerals. Currently, oxidized minerals are being depleted as the depth of mining increases, which means that many hydrometallurgical plants will become obsolete in the near future. Furthermore, chalcopyrite ($CuFeS_2$) is the most common copper ore. The extraction of copper from this mineral is carried out by means of pyrometallurgical processing of copper concentrates. Hydrometallurgy is an alternative route, which has some important advantages such as the ability to treat low-grade minerals, easier control of by-products to limit the generation of impurities and obtain greater copper extraction, lower investment and operation costs [2]. Such a process has already been used for the heap leaching of copper from secondary sulfides [3], but it is not yet possible to properly apply this approach for

chalcopyrite minerals on an industrial scale [4]. This is because chalcopyrite is refractory, causing slow leaching kinetics due to its strong passivation in acid solutions [4–7].

It should be noted that hydrometallurgical processes use large volumes of water. Hydrometallurgical processes consumed a total of 1.74 m$^3$/s of fresh water during 2018 in Chile [8]. There continue to be a decrease in water resources in the world, due to climate change, the exponential growth of the human population and the growth of industries. Therefore, due to the depletion of drinking water and the challenge of leaching chalcopyrite, it is necessary to look for new strategies to develop an effective copper leaching process, which is environmentally friendly and inexpensive. For this reason, several investigations have used chlorinated media, which gives better results than the leaching of sulfide minerals in sulfate medium. This chemical element is an essential component of seawater (approximately 20 g/L of chlorine). The presence of chlorides in a medium with sulfuric acid provides better leaching kinetics while chloro complex formation has an impact on redox potential [9–16]. Furthermore, chloride media can stabilize cuprous ions and increase the solubility of copper. It has also been shown that chloride ions can modify the layer of particle leached, which is commonly elemental sulfur, making it more porous and promote copper dissolution [10,14,17–19]. Acid leaching of chalcopyrite requires oxidants and some authors have investigated the use of nitrate and nitrite salts ($KNO_3$, $NaNO_3$ and $NaNO_2$) [20–22]. These chemical species have a higher oxidative potential compared to other common oxidants such as ferric ion [22–24]. Sodium nitrate is very abundant in northern Chile and is obtained by processing caliche [25–27]. It is obtained by leaching caliche with water or seawater and then crystallizing the solution by cooling or evaporation. After evaporation has been employed, the waste salts (tails) from the solar ponds still contain a significant amount (4.6% of $NaNO_3$) of nitrate salts [28]. These discarded salts can be used as an oxidant for chalcopyrite leaching. The use of these nitrogenous species has several positive effects; they have the ability to partially or completely oxidize the sulfide contained forming elemental sulfur, the reaction speed is faster, reactors with less volume are needed, high temperatures or pressures are not necessary and the oxidation-reduction potential of the reaction is high. [21,24,29–33]. Narangarav et al. [31] studied the leaching of chalcopyrite concentrate in an acid-nitrate media. They determined that copper extraction can reach over 85% by increasing the nitrate concentration from 0.2 to 1 M, and that the reaction has an activation energy of 15.96 kJ/mol indicating that the kinetics are chemically controlled. Sokić et al. [21] studied the leaching of chalcopyrite concentrate using a solution with sodium nitrate and sulfuric acid. Oxidation of the mineral can be achieved in two ways: the first where the $NO_3^-$ ion is the oxidant that is subsequently reduced to NO or $NO_2$ and second, the oxygen obtained during the decomposition of $NO_3^-$, also acts as oxidant. The authors proposed that the dissolution of chalcopyrite using nitrate as an oxidizing agent in an acid solution forms an elemental sulfur product, according to the following equation:

$$CuFeS_2 + 5NaNO_3 + 5H_2SO_4 \rightarrow CuSO_4 + 0.5Fe_2(SO_4)_3 + 2.5Na_2SO_4 + 2S^0 + 5NO_2 + 5H_2O \quad (1)$$

The process is temperature dependent, with better copper extractions obtained at higher temperatures, the activation energy of the reaction was 83 kJ/mol. The authors demonstrated that elemental sulfur is formed on the mineral surface through optical microscopy (SEM). Hernandez et al. [20] investigated the stirred leaching of low copper sulfide grade ore in a nitrate-chloride-acid media. An extraction of 80% Cu was obtained at 45 °C in 7 days. The presence of chloride in the leaching solution increased the copper dissolution. More recently, Hernandez et al. [34] investigated the effect of agglomeration and curing on the leaching of copper sulfide in acid-nitrate-chloride media. The maximum copper extraction of 58.6% was achieved after 30 days of curing at 45 °C.

In this study, samples of chalcopyrite were leached using nitrate-chloride-acid media. The source of chloride was provided by seawater or brine discarded from the reverse osmosis process. The effect of sulfuric acid, nitrate and chloride concentration, source of water (dissolvent), temperature, solid/liquid ratio, particle size, mineral sample, and pretreatment were studied.

## 2. Materials and Methods

### 2.1. Ore Samples

Five chalcopyrite samples were obtained from different copper mines of northern Chile. The samples were milled by hand until the desired particle size (−250 + 150 μm, −150 + 75 μm, and −75 μm) was achieved. Chemical analyses of the samples (see Table 1) were determined by atomic absorption spectrometry (AAS, Perkin–Elmer 2380, Perkin–Elmer, Wellesley, MA, USA) and the mineralogy (see Table 2) was determined by quantitative X-ray diffraction (Siemens/Bruker, Semi-QXRD, model D8 Advance, Karlsruhe, Germany). This equipment provides a semi-quantitative result using TOPAS (total pattern analysis software) (version 5.0, Bruker-AXS, Karlsruhe, Germany) to quantify the mineralogy of the sample.

**Table 1.** Chemical analyses of the chalcopyrite samples.

| Sample | Cu (wt.%) | Fe (wt.%) |
|--------|-----------|-----------|
| A | 29.9 | 25.6 |
| B | 25.6 | 32.7 |
| C | 18.8 | 28.1 |
| D | 23.6 | 33.1 |
| E | 21.2 | 19.6 |

**Table 2.** Mineralogy of samples used (particle size of −150 + 75 μm).

| Chemical Formula | Mineral | % | | | | |
|------------------|---------|------|------|------|------|------|
| | | A | B | C | D | E |
| $CuFeS_2$ | Chalcopyrite | 85 | 90 | 87 | 89 | 45 |
| $FeS_2$ | Pyrite | 2.1 | 8.1 | 11.3 | 11.2 | - |
| $SiO_2$ | Quartz | 9.2 | 2.4 | 1.6 | - | 28.9 |
| $K_{0.4}Na_{0.6}Cl$ | Potassium-halite | 3.5 | - | - | - | - |
| $NaAlSi_3O_8$ | Albite | - | - | - | 0.1 | - |
| $CaSO_4$ | Anhydrite | - | - | - | - | 14.8 |
| $CaSO_4 \cdot 2H_2O$ | Gypsum | - | - | - | - | 7.2 |
| $KAl_2(Si_3Al)O_{10}(OH)_2$ | Muscovite | - | - | - | - | 4.1 |
| | Total | 100 | 100 | 100 | 100 | 100 |

### 2.2. Leaching Solutions

Leaching solutions were prepared using sulfuric acid ($H_2SO_4$, 95 to 97%), sodium chloride (NaCl, 99.5%) and sodium nitrate ($NaNO_3$ 99.5%). All reagents are analytical grade. Various dissolvents, distilled water, seawater, and brine from reverse osmosis were used. Seawater was obtained from San Jorge Bay, Antofagasta Chile. It was pumped and filtered through a 1 μm polyethylene membrane. Brine was obtained from Coloso's desalination plant, Antofagasta, Chile. A comparison of the composition of seawater and brine is shown in Table 3.

**Table 3.** Compositions of seawater and brine used in this study (mg/L).

| Dissolvent | Ionic Species | | | | | | | | |
|------------|------|------|------------|------------|------------|------|--------------|------------|--------------|
| | $Na^+$ | $K^+$ | $Mg^{2+}$ | $Ca^{2+}$ | $Cu^{2+}$ | $Cl^-$ | $SO_4^{2-}$ | $NO_3^-$ | $HCO_3^-$ |
| Seawater | 11,250 | 401 | 1256 | 427 | <0.1 | 20,289 | 2758 | 2.4 | 149 |
| Brine | 19,768 | 746 | 2297 | 355 | 0.1 | 36,074 | 5063 | 6.4 | 236 |

## 2.3. Experimental Procedure

### 2.3.1. Leaching Tests

Stirred leaching was carried out with jacketed glass reactors of 1 L. Water at desired temperature was pumped through the jacket of the reactors. The temperature of water was maintained with a thermostatic bath (Julabo bath F25-ME Refrigerated/Heating Circulator, JULABO GmbH, Germany). Mechanical stirring with a Teflon bar was used to provide agitation of the pulp (450 rpm). The leaching solution (500 mL) was introduced in the reactor. When the leaching solution reach the desired temperature, chalcopyrite sample was added into the reactor and the stirring started. At certain times, samples of leaching solutions were removed from the system to be analyzed for copper and iron (AAS). Moreover, the pH and redox potential (ORP) were measured (Hanna portable pH/ORP meter, model HI991003, accuracy ±0.02 pH y ± 2 mV, Ag/AgCl reference electrode). When the leaching time was achieved, the pulp was filtered. The solid was washed with distilled water and dried at 60 °C for 24 h. Solid residue was analyzed by optical microscopy with reflected light, and by scanning electron microscopy (SEM-EDX, JEOL 6260 LV, Tokyo, Japan). The final leaching solutions was analyzed for Cu and Fe (AAS). Copper and iron extractions were calculated using the grades of the head and residual ore, which was corroborated with the results obtained for the leaching solutions. A standard deviation of ±2% was obtained by calculating the copper/iron extraction rates from solid residue and leaching solution for all the tests.

Table 4 shown the leaching tests carried out. The variables studied were: sulfuric acid concentration (0.1, 0.25, 0.5, 0.7 and 1 M), sodium nitrate concentration (0, 0.1, 0.25, 0.5, 0.7 and 1 M), chloride concentration (0, 20, 36, 60 g/L), dissolvent (distilled water, seawater, and brine), temperature (45, 55 and 65 °C), solid/liquid ratio (2, 5 and 10 g/L), particle sizes (−250 + 150 μm, −150 + 75 μm, and −75 μm), chalcopyrite sample (A, B, C, D, and E) and pretreatment (with and without).

**Table 4.** Leaching test parameters with metal extraction results.

| N° | $H_2SO_4$ (M) | $NaNO_3$ (M) | T (°C) | S/L Ratio (g/L) | Size (μm) | Time (h) | Dissolvent | Sample | Other | Cu (%) | Fe (%) |
|---|---|---|---|---|---|---|---|---|---|---|---|
| 1 | 0.5 | 0.5 | 45 | 2 | −150 + 75 | 168 | seawater | A | - | 10.1 | 6.7 |
| 2 | 0.1 | 0.1 | 45 | 2 | −150 + 75 | 168 | seawater | A | - | 4.3 | 3.4 |
| 3 | 1.0 | 1.0 | 45 | 2 | −150 + 75 | 288 | seawater | A | - | 26.3 | 24.3 |
| 4 | 0.7 | 0.7 | 45 | 2 | −150 + 75 | 432 | distilledwater | A | - | 7.3 | 9.5 |
| 5 | 0.7 | 0.7 | 45 | 2 | −150 + 75 | 432 | seawater | A | - | 26.8 | 18.8 |
| 6 | 0.5 | 0.5 | 45 | 2 | −150 + 75 | 432 | seawater | A | - | 16.0 | 11.96 |
| 7 | 0.7 | 0.7 | 45 | 2 | −250 + 150 | 432 | seawater | A | - | 6.7 | 4.2 |
| 8 | 0.7 | 0.7 | 45 | 2 | −75 | 432 | seawater | A | - | 55.2 | 43.1 |
| 9 | 0.7 | 0.7 | 65 | 2 | −150 + 75 | 432 | seawater | A | - | 92.3 | 88.9 |
| 10 | 0.7 | 0.7 | 55 | 2 | −150 + 75 | 432 | seawater | A | - | 60.0 | 56.4 |
| 11 | 0.7 | 0.7 | 45 | 5 | −150 + 75 | 432 | seawater | A | - | 55.4 | 49.7 |
| 12 | 0.7 | 0.7 | 45 | 10 | −150 + 75 | 432 | seawater | A | - | 78.8 | 73.5 |
| 13 | 0.7 | 0.7 | 45 | 2 | −150 + 75 | 432 | seawater | A | Pretreatment * | 55.9 | 54.9 |
| 14 | 0.7 | 0.7 | 45 | 2 | −150 + 75 | 432 | seawater | A | Pretreatment ** | 64.7 | 57.0 |
| 15 | 0.7 | 0.7 | 45 | 2 | −150 + 75 | 432 | seawater | A | $[Cl^-]$ = 60 g/L | 64.4 | 73.5 |
| 16 | 0.7 | 0.7 | 45 | 2 | −150 + 75 | 432 | brine | A | - | 49.9 | 64.8 |
| 17 | 0.7 | 0.7 | 45 | 2 | −150 + 75 | 432 | seawater | B | - | 46.5 | 42.7 |
| 18 | 0.7 | 0.7 | 45 | 2 | −150 + 75 | 432 | seawater | C | - | 27.2 | 33.4 |
| 19 | 0.25 | 0.7 | 45 | 2 | −150 + 75 | 432 | seawater | A | - | 15.7 | 13.4 |
| 20 | 0.5 | 0.7 | 45 | 2 | −150 + 75 | 432 | seawater | A | - | 17.8 | 14.4 |
| 21 | 0.7 | 0.7 | 45 | 2 | −150 + 75 | 432 | seawater | D | - | 27.97 | 53.3 |
| 22 | 0.7 | 0.7 | 45 | 2 | −150 + 75 | 432 | seawater | E | - | 67.4 | 68.5 |
| 23 | 0.7 | 0.0 | 45 | 2 | −150 + 75 | 432 | seawater | A | - | 4.7 | 4.1 |
| 24 | 0.7 | 0.25 | 45 | 2 | −150 + 75 | 432 | seawater | A | - | 35.1 | 45.9 |
| 25 | 0.7 | 0.5 | 45 | 2 | −150 + 75 | 432 | seawater | A | - | 45.9 | 67.2 |

* Pretreatment conditions: 1.15 g $NaNO_3$, 0.76 mL $H_2SO_4$ and 1 g of seawater (values based on $\frac{1}{2}$ of stoichiometry, equation 1). ** Pretreatment conditions: 0.58 g $NaNO_3$, 0.38 mL $H_2SO_4$, 1 g of seawater (values based on $\frac{1}{4}$ of stoichiometry, Equation (1)).

### 2.3.2. Pretreatment Tests

Solid sample was placed on a Petri dish. Solid NaCl and NaNO$_3$ were added and mixed with a spatula. Drops of concentrated sulfuric acid and seawater (added with sprinkler) were added and mixed to give a homogeneous paste with moisture content of 15%. The Petri dish was sealed using parafilm and left at room temperature for 3 days. Then, the pretreated solid was leached using the procedure detailed previously.

## 3. Results and Discussions

Copper and iron extractions and the corresponding leaching conditions used are shown in Table 4.

### 3.1. Effect of Different Variables

#### 3.1.1. Effect of Temperature

Figure 1a shows the effect of temperature on copper extraction. An increase of temperature produces an increase of copper extraction. The best copper extraction was achieved at 65 °C after 432 h (92.3%). After 120 h, the copper extractions achieved at 45 and 65 °C were 10.8% and 81.9%, respectively. This larger difference is maintained when the leaching time was 432 h, where copper extraction achieved at 45 and 65 °C were 26.8% and 92.3%, respectively. These results are consistent with others reported in the literature [21,30]. The pH values were near 0 during the tests, due to high acidity. ORP values were in a range of 770 and 820 mV vs Ag/AgCl for all samples. It was observed that an increase of temperature produces an increase of ORP potential. This is explained by the formation of chloro complexes in the presence of a greater amount of copper and iron present in the solution, as these complexes are highly oxidizing [11,35–39]. Figure 1b shows the molar ratios between dissolved copper and iron during the leaching time. During the leaching time, the molar ratios are close to 1:1, which indicates that both chemical elements are being dissolved in parallel which is expected from the molecular formula (CuFeS$_2$). Similar plots are shown for all experiments in this manuscript.

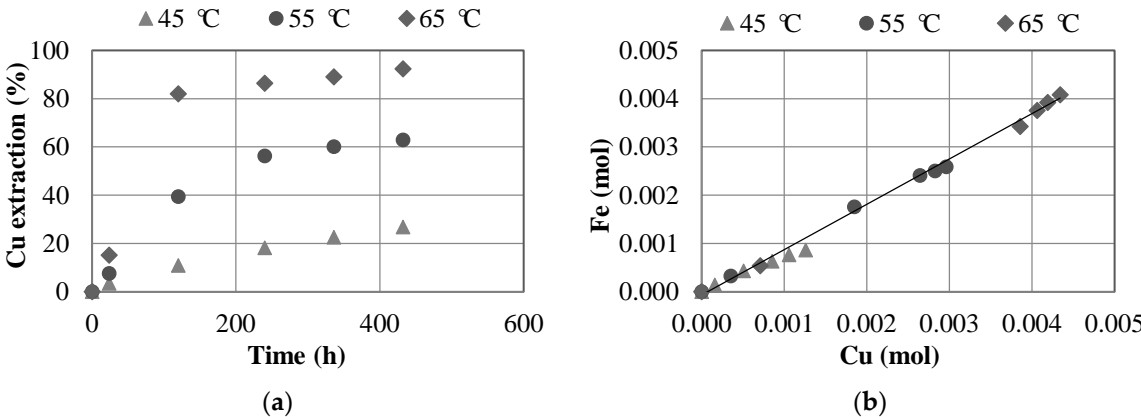

**Figure 1.** (**a**) Copper extraction (%) versus time (h) at different temperatures: 45 °C, 55 °C and 65 °C. (**b**) Cu/Fe molar ratio at different temperatures: 45, 55 and 65 °C. Experimental conditions: [NaNO$_3$] = 0.7 M, [H$_2$SO$_4$] = 0.7 M, seawater as dissolvent, −150 + 75 μm, S/L ratio 2 g/L, 350 rpm, sample A.

#### 3.1.2. Effect of Solid/Liquid Ratio

Figure 2a shows the effect of solid/liquid ratio on copper extraction. The maximum copper extractions achieved were 26.8%, 55.4% and 85.2% for solid/liquid ratios of 2, 5 and 10 g/L, respectively. It can be observed that when solid/liquid ratio increased, the dissolution of copper and iron increased. The pH values were near 0 during the tests. ORP values were in a range of 780 and 800 mV vs Ag/AgCl. At these potentials it is expected that ferric ion is present and that this will facilitate chalcopyrite dissolution. Figure 2b shows that the during the leaching time, the molar ratios are close to 1:1.

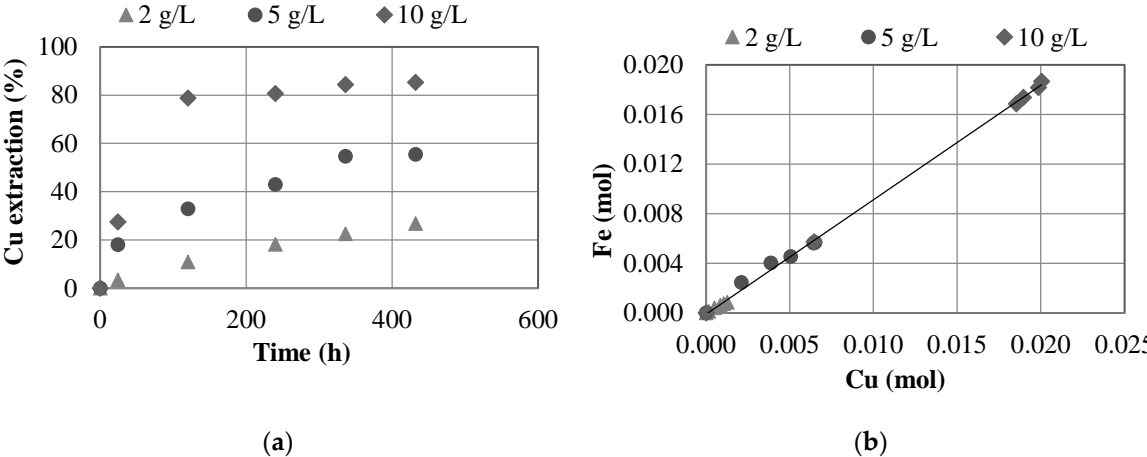

(a)　　　　　　　　　　　　　　　　　(b)

**Figure 2.** (**a**) Copper extraction (%) versus time (h) at different solid/liquid ratio: 2, 5 and 10 g/L. (**b**) Cu/Fe molar ratio at solid/liquid ratio: 2, 5 and 10 g/L. Experimental conditions: [NaNO$_3$] = 0.7 M, [H$_2$SO$_4$] = 0.7 M, seawater as dissolvent, −150 + 75 μm, 45 °C, 350 rpm, sample A.

### 3.1.3. Effect of Dissolvent (Process Water)

Figure 3a shows the effect of dissolvent used on copper extraction. The copper extraction obtained after 432 h reached 50.3, 26.8 and 7.3% when brine, seawater, and distilled water were used, respectively. The best extraction was obtained when brine was used. This shows that a high concentration of chloride ion improves the dissolution of chalcopyrite. When seawater was used as a dissolvent, copper extraction was 3.6 times greater compared to when distilled water was used. When brine from a desalination plant was used, copper extraction was approximately 7 times greater than when distilled water was used. Therefore, the greater the amount of chloride, the greater the extraction of copper and iron from the chalcopyrite. This result demonstrates the advantage of using discard brine from the reverse osmosis process, which is not currently used and is returned to the sea. The presence of the chloride ion (Cl$^-$) can form chloride complexes with metals such as copper and iron present in the chalcopyrite solution. These chloride complexes are oxidants that improve the efficiency of the process [15,16,40]. Furthermore, it has been shown that the presence of chloride in a leaching system affects the mineral surface making the outer sulfur layer in contact with the solution porous, which allows the dissolution reaction to continue [16,17]. The pH values were near 0 during the tests. ORP values were in a range of 713 and 797 mV vs Ag/AgCl. A higher redox potential was obtained when seawater and brine were used instead of distilled water. Velásquez-Yévenes et al. [41] have also determined that the potential of the leaching solution increases with the increasing of chloride concentration. Yoo et al. [42] determined that the formation of chloro complexes of copper increases with increasing chloride concentration, affecting the redox potential (increase). Figure 3b shows that the molar ratios are close to 1:1 during the tests.

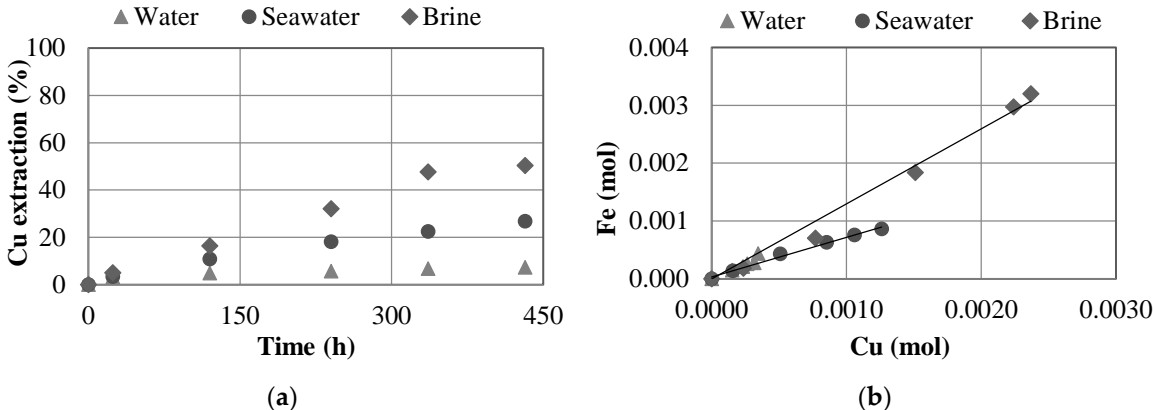

**Figure 3.** (**a**) Copper extraction (%) versus time (h) when different dissolvent were used: distilled water (water), seawater, and brine. (**b**) Cu/Fe molar ratio when different dissolvent were used: distilled water (water), seawater, and brine. Experimental conditions: $[NaNO_3]$ = 0.7 M, $[H_2SO_4]$ = 0.7 M, $-150 + 75$ µm, S/L ratio 2 g/L, 45 °C, 350 rpm, sample A.

### 3.1.4. Effect of Particle Size

Figure 4a shows the effect of particle size on copper extraction. The copper extraction obtained in 432 h reached 6.7, 26.8 and 55.2% when granulometries between $-250 + 150$ µm, $-150 + 75$ µm and $-75$ µm respectively, were used. This is expected as finer particles have a larger contact surface area; therefore, a higher dissolution is obtained compared to coarse particles [21,43–45]. A smaller particle size results in a larger copper and iron extraction, where extraction from a particle size of $-75$ µm in just 1 day exceeded the maximum copper and iron extraction for a size between $-250 + 150$ µm, and in 5 days, it exceeded the maximum extraction reached for a size between $-150 + 75$ µm. The decrease in particle size has positive consequences: shorter leaching times, better leaching kinetics, but also negative, such as increased costs and energy for sample grinding. An approach to make use of this benefit would be to leach copper concentrates due to their low particle sizes. The pH values were near 0 during the tests. ORP values were in a range of 740 and 804 mV vs Ag/AgCl. A higher redox potential was obtained when fine particles size ($-75$ µm) were leached in comparison with bigger sizes ($-250 + 150$ µm or $-150 + 75$ µm) and this can be attributed to the higher concentrations of cupric and ferric ions. Figure 4b shows the molar ratios during the leaching time. In these tests, copper dissolutions were major to iron dissolution. This trend increases when the particle size decrease.

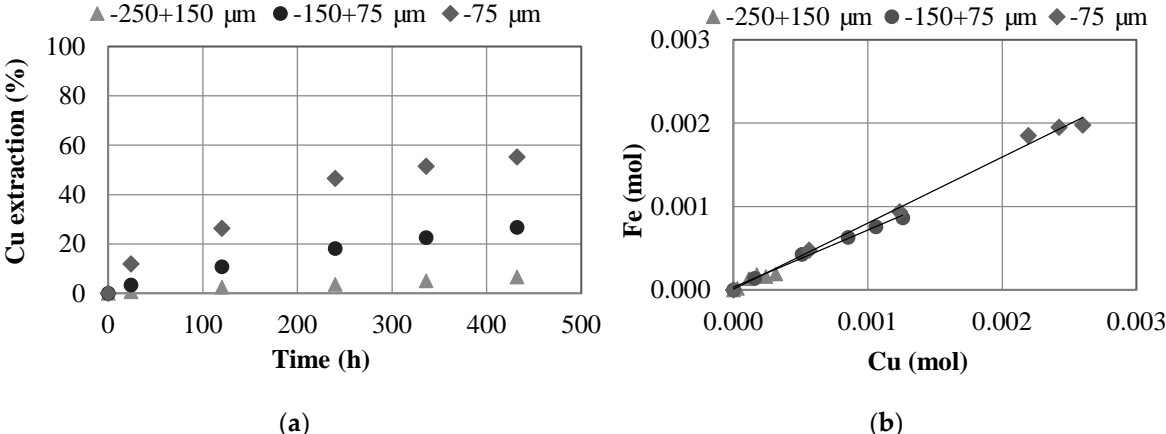

**Figure 4.** (**a**) Copper extraction (%) versus time (h) at different particle size: $-250 + 150$ µm, $-150 + 75$ µm and $-75$ µm. (**b**) Cu/Fe molar ratio at different particle size: $-250 + 150$ µm, $-150 + 75$ µm and $-75$ µm. Experimental conditions: $[NaNO_3]$ = 0.7 M, $[H_2SO_4]$ = 0.7 M, seawater as dissolvent, S/L ratio 2 g/L, 45 °C, 350 rpm, sample A.

### 3.1.5. Effect of Nitrate Concentration

Figure 5a shows the effect of nitrate concentration on copper extraction. The tests gave 4.7%, 35.1%, 49.1% and 26.8% copper extractions at 0, 0.25, 0.5 and 0.7 M of $NaNO_3$, respectively after 432 h. This confirms that as the concentration of sodium nitrate increases, the extraction of copper increases. This reagent favors the oxidation of copper to cupric ion. It should be noted that when a concentration of 0.7 M is used, the extraction decreases, which can be attributed to the high ionic strength present in the leaching media due to the presence of ions from seawater, nitrate, and the acid added to the system. This effect has been observed in other systems (Figure 4 in [36], Figure 2 in [20]). Different researchers indicate that a minimum of 0.15 M of $NaNO_3$ should be used to see positive changes in copper extraction when used as an oxidizing agent in leaching. If more of this reagent is added, natrojarosite $(Na(Fe(OH)_2)_3(SO_4)_2)$ can be generated in the reaction and these product can remain on the surface of the mineral preventing a continuous dissolution of copper and iron [4,16,17,46–48]. A positive effect was observed when the test without the addition of sodium nitrate ($\approx$440 mV were ORP values during the test) is compared with the test with 0.25 M sodium nitrate ($\approx$700 mV were ORP values during the test). A significant change in copper extraction was observed when sodium nitrate is added confirming the oxidizing power of this reagent. Figure 5b shows the molar ratios of the tests. At 0.5 M $NaNO_3$, the molar dissolution of iron was higher than the molar dissolution of copper. This could be explained because the dissolution of pyrite present in the sample, would help to provide iron to the leaching media, helping the chalcopyrite dissolution due to presence of ferric ion at high redox potential of the solution.

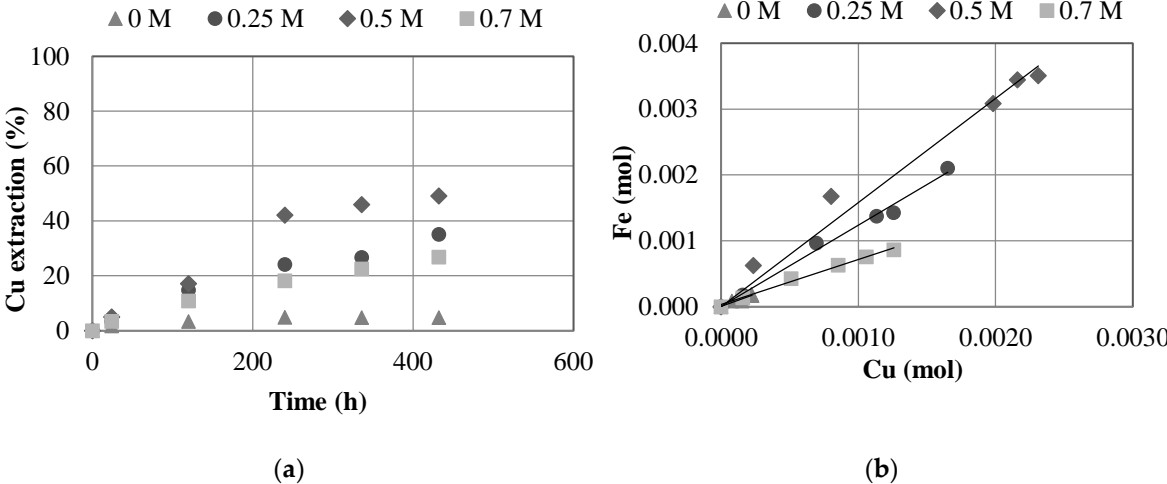

(**a**)　　　　　　　　　　　　　　　　　(**b**)

**Figure 5.** (**a**) Copper extraction (%) versus time (h) at different sodium nitrate concentration: 0, 0.25, 0.5 and 0.7 M. (**b**) Cu/Fe molar ratio at different sodium nitrate concentration: 0, 0.25, 0.5 and 0.7 M. Experimental conditions: $[H_2SO_4]$ = 0.7 M, seawater as dissolvent, −150 + 75 μm, S/L ratio 2 g/L, 45 °C, 350 rpm, sample A.

### 3.1.6. Effect of Sulfuric Acid Concentration

Figure 6a shows the effect of sulfuric acid concentration on copper extraction. The tests gave copper extraction of 15.7, 17.8 and 26.8% at 0.25, 0.5 and 0.7 M $H_2SO_4$, respectively after 432 h. Tests at 0.25 and 0.5 M $H_2SO_4$, did not show a significant difference in copper extraction. At 0.7 M $H_2SO_4$ there is a greater increase compared to tests with lower acid concentration. At a higher concentration of sulfuric acid, the oxidation-reduction potential was greater, increasing from 716–730 mV (0.25 M) to 770–790 mV (0.7 M). According to Sokić et al., [21] the oxidizing potential of nitrate ions is favored by the increase of acid concentration. This was also observed by Narangarav et al. [31]. The final pH values were maintained near 0 in 0.7 M $H_2SO_4$, 0.3 in 0.5 M $H_2SO_4$ and 1 in 0.25 M $H_2SO_4$. Figure 6b shows that almost all tests show correlations close to 1:1.

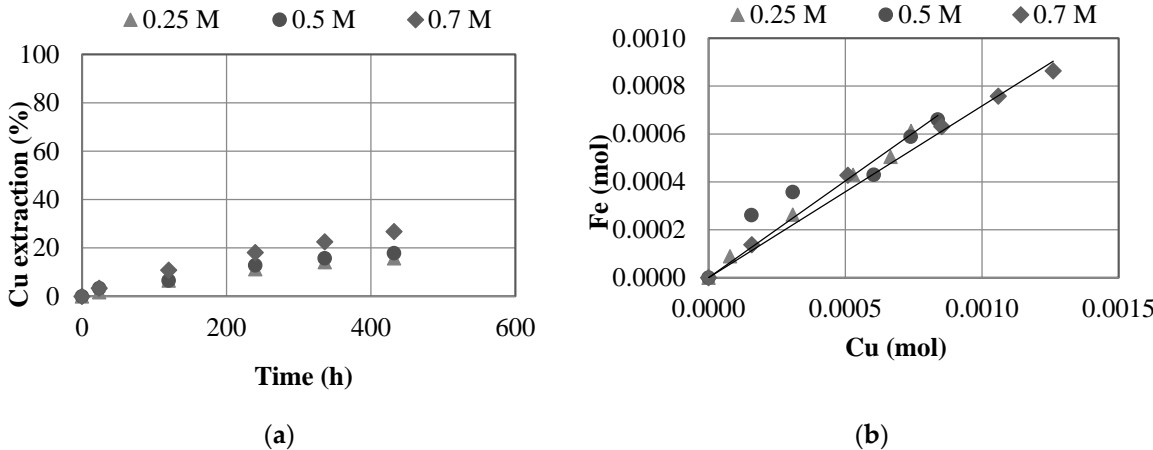

**Figure 6.** (**a**) Copper extraction (%) versus time (h) at different sulfuric acid concentration: 0.25, 0.5 and 0.7 M. (**b**) Cu/Fe molar ratio at different sulfuric acid concentration: 0.25, 0.5 and 0.7 M. Experimental conditions: [NaNO$_3$] = 0.7 M, seawater as dissolvent, −150 + 75 μm, S/L ratio 2 g/L, 45 °C, 350 rpm, sample A.

### 3.1.7. Effect of Chloride Concentration

Figure 7a shows the effect of chloride concentration on copper extraction. The tests gave copper extractions of 7.3%, 26.8%, 50.3%, and 64.4% for 0, 20, 36 and 60 g/L chloride, respectively, after 432 h. The increase of chloride concentration increases the copper and iron extractions. The positive effect of increase of chloride ions in the leaching solution has been studied by several researchers [12,14,42]. The presence of chloride changes the morphology of the mineral surface enabling a porous sulfur layer and increase the solubility of the metals in these tests [14,39,41]. Redox potential increased when the chloride concentration increased from 710 mV at 0 g/L chloride to 800 mV at 60 g/L chloride. Moreover, final pH values at 0 g/L chloride were near 0.3 whereas the pH values when chloride ions were in the media were near 0 during all the tests. This can be attributed to the increase proton (H$^+$) activity when chloride ions are present in the media [49]. It should also be noted that the use of brine showed beneficial results for the dissolution of copper and iron. Figure 7b shows that at 20 g/L Cl$^-$, the copper dissolution is higher than the iron dissolution. This trend is different from 36 and 60 g/L Cl$^-$, where the behavior is opposite, probably due to the increase of chloride in the leaching medium that helps the pyrite dissolution.

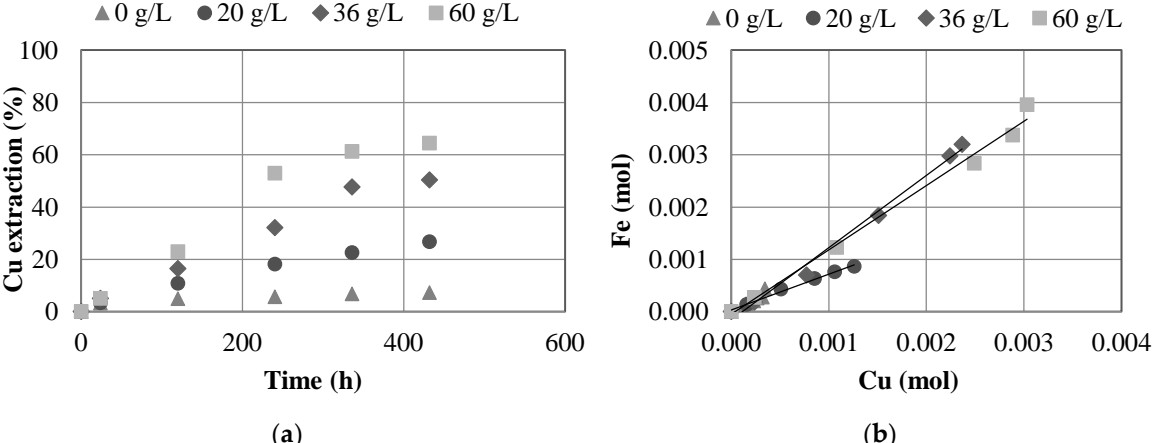

**Figure 7.** (**a**) Copper extraction (%) versus time (h) at different chloride concentration: 0, 20, 36 and 60 g/L. (**b**) Cu/Fe molar ratio at different chloride concentration: 0, 20, 36 and 60 g/L. Experimental conditions: [NaNO$_3$] = 0.7 M, [H$_2$SO$_4$] = 0.7 M, −150 + 75 μm, S/L ratio 2 g/L, 45 °C, 350 rpm, sample A.

Figure 8 shows the surface of mineral A leached in a solution with 36 g/L of chloride (Figure 8a, Test 16) and 60 g/L of chloride (Figure 8b, Test 15) at concentrations of 0.7 M $H_2SO_4$ and 0.7 M $NaNO_3$. The images show an amorphous surface with many reliefs. EDS analysis shows the peaks for elemental sulfur at the residue surface. Both samples are mainly covered by sulfur, which confirms the results of other studies [16,21,50].

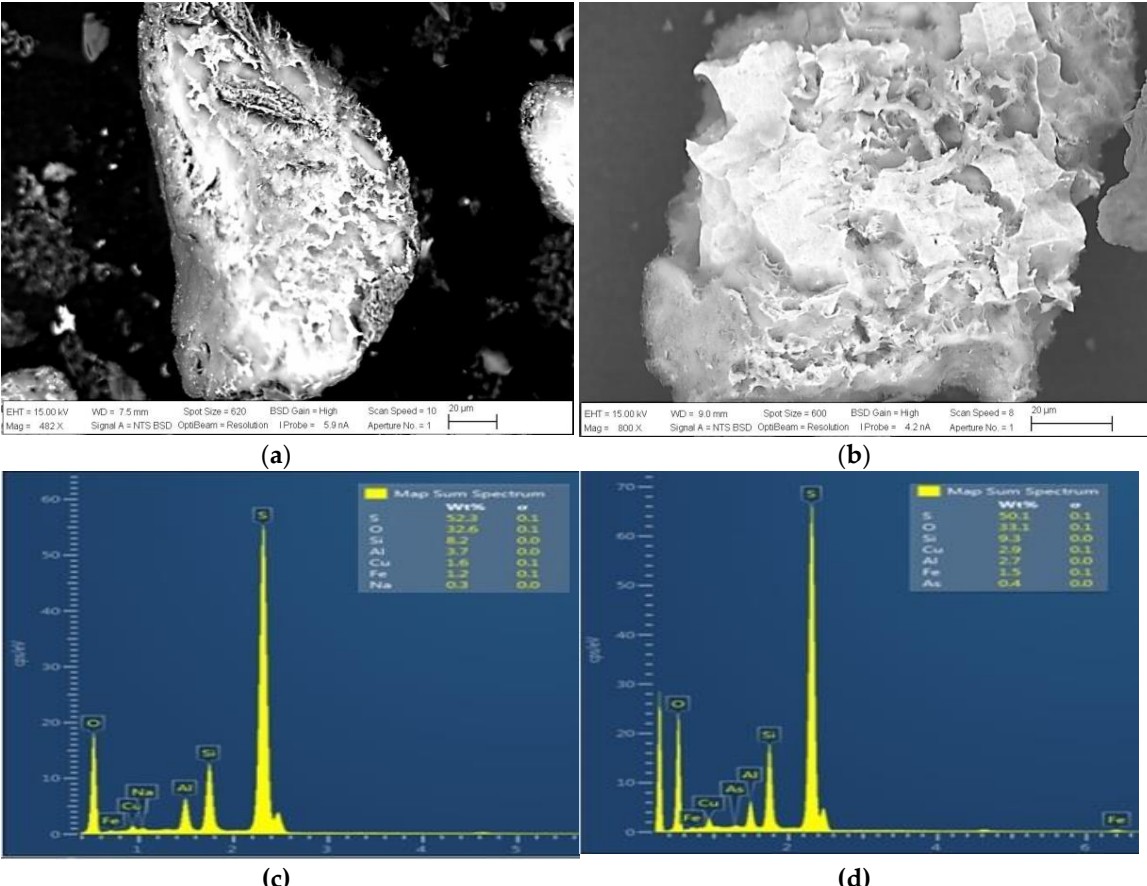

**Figure 8.** (**a**) SEM micrographs of chalcopyrite leached with 36 g/L chloride, Test 16; (**b**) SEM micrographs of chalcopyrite leached with 60 g/L chloride, Test 15; (**c**) EDS spectra of chalcopyrite leached with 36 g/L chloride, Test 16 (mapping) and (**d**) EDS spectra of chalcopyrite leached with 60 g/L chloride, Test 15 (mapping). Experimental conditions: $[NaNO_3]$ = 0.7 M, $[H_2SO_4]$ = 0.7 M, $-150 + 75$ μm, S/L ratio 2 g/L, 45 °C, 350 rpm, sample A.

### 3.1.8. Effect of Variation of Nitrate and Acid Concentration

Figure 9a shows the effect of variation of both reagents, nitrate and sulfuric acid concentration, on copper extraction. Increased $H_2SO_4$ and $NaNO_3$ concentrations have a positive effect on the copper and iron dissolution. A low concentration of reagents (0.1 M) produces a low copper extraction which does not exceed 5% Cu after 168 h. At 1 M concentration the extraction, in just 12 days exceeds 25% Cu and 24% Fe. During the test, redox potential values were near 700 mV at 0.1 M of both reagents, 730 mV at 0.5 M, 770 mV at 0.7 M and 835 mV at 1 M. The final pH values varied from 1 at 0.1 M, 0.3 at 0.5 M, 0.1 at 0.7 M and 0 at 1M of both reagents. Figure 9b shows that the molar ratios are close to 1:1 during the test performed at 1 M of both reagents. At 0.1, 0.5 and 0.7 M, the copper dissolution was higher than iron dissolution.

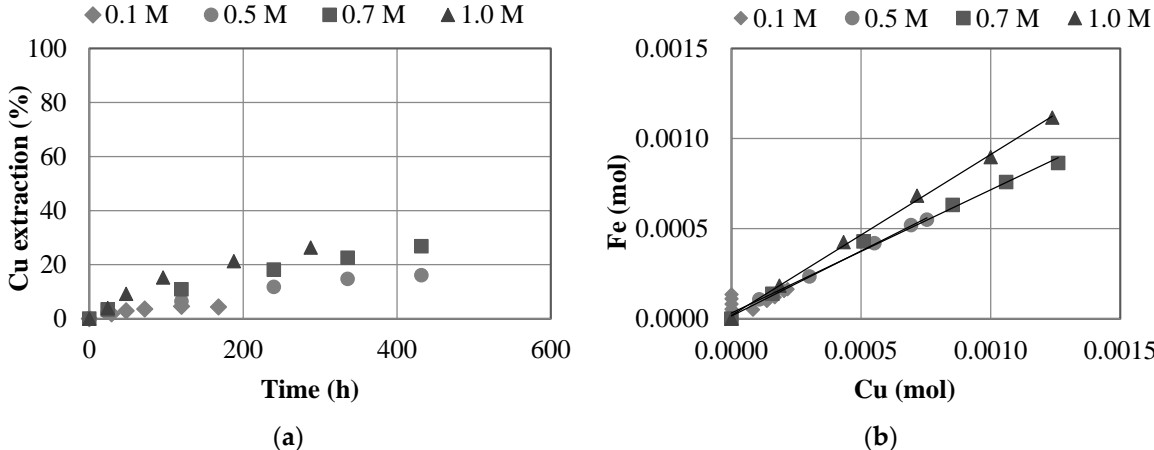

**Figure 9.** (**a**) Copper extraction (%) versus time (h) at different equivalent nitrate and acid concentrations: [NaNO$_3$] = [H$_2$SO$_4$] = 0.1, 0.5, 0.7 and 1.0 M. (**b**) Cu/Fe molar ratio at different equivalent nitrate and acid concentrations: [NaNO$_3$] = [H$_2$SO$_4$] = 0.1, 0.5, 0.7 and 1.0 M. Experimental conditions: seawater as dissolvent, −150 + 75 μm, S/L ratio 2 g/L, 45 °C, 350 rpm, sample A.

### 3.1.9. Effect of Chalcopyrite Grade

For these tests, five different chalcopyrite samples were used. The chemical composition of each mineral is shown in Table 2. Results obtained can be classified into two groups as shown in Figure 10a. Copper extractions for chalcopyrite A, C, and D have an average value of 27.3% ± 0.6% and samples B and E have an average of 64.1% ± 4.6%. As shown in Figure 10a, the copper extraction begins to decrease after 240 h, causing an asymptotic curve. Samples A, B, C, and D have a similar percentage of chalcopyrite (approximately 85%) compared to chalcopyrite E (close to 45%). The differences observed in both groups can be attributed to the degree of chalcopyrite release in the rock, where its presence could be disseminated, occluded and/or free in the particle [51,52]. Therefore, the release of this mineral could be similar in samples B and D with a higher degree of release compared to samples A, C, and D. Further characterization studies are required to determine the liberation of chalcopyrite in these samples. On the other hand, from the comparison of the particle size distributions of the chalcopyrite samples, it can be seen (Figure 10c) that the accumulated percentages for samples B and E are similar, with P$_{80}$ values of 171.8 and 171.7 μm, respectively, and quite different from the distributions obtained for samples A, C, and D, for which the P$_{80}$ values are 148.0, 149.6 and 160.3 μm, respectively. These differences are expected to influence the leaching behavior observed in the tests carried out. According to the results shown in Figure 4a, the effect of particle size on leaching is significant, and which is also demonstrated in these tests.

The ORP values during the tests were in a range of 753–800 mV. The pH values were low 0.3, during the leaching tests.

Figure 10b shows molar ratios. The behaviors of the values are different in all tests, probably because the samples have different mineralogy that affect the results of copper and iron dissolution over the time.

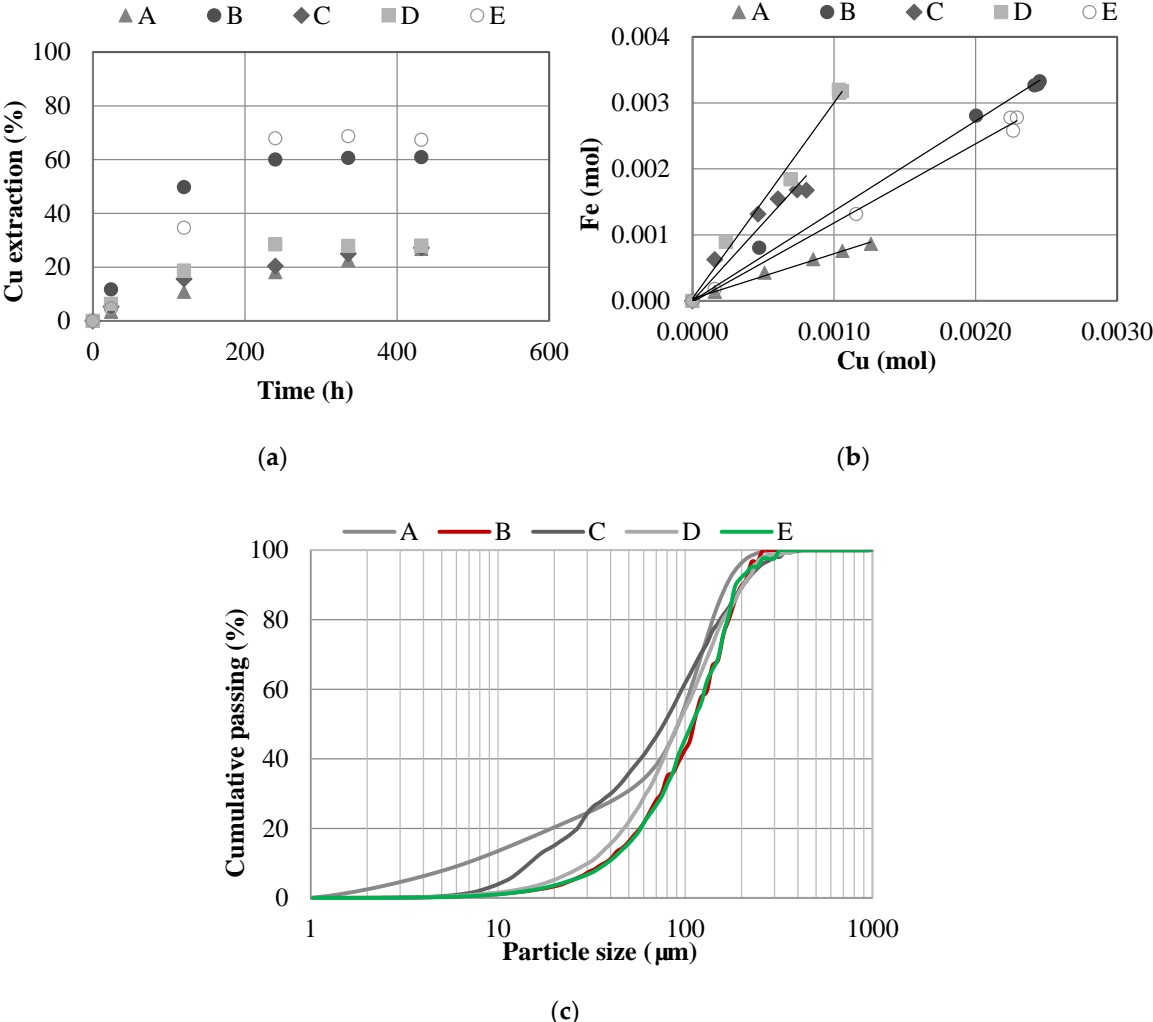

**Figure 10.** (**a**) Copper extraction (%) versus time (h) when different chalcopyrite samples were used: A, B, C, D, and E. (**b**) Cu/Fe molar ratio when different chalcopyrite samples were used: A, B, C, D, and E. (**c**) Particle size distribution of samples: A, B, C, D, and E after the experiments. Experimental conditions: [NaNO$_3$] = 0.7 M, [H$_2$SO$_4$] = 0.7 M, seawater as dissolvent, −150 + 75 μm, S/L ratio 2 g/L, 45 °C, 350 rpm.

### 3.1.10. Effect of Pretreatment

Two samples of chalcopyrite A were pretreated at conditions shown in Table 4, Tests 13, and 14. When pretreatment was concluded (3 days), the solid was washed using freshwater and the copper extraction was obtained (AAS). The pretreatment achieved 47.5 and 35.8% Cu extractions in Test 13 and 14, respectively. This demonstrated the efficiency of the process. Curing (pretreatment) with reagents followed by a rest time produces an acceleration of the dissolution kinetics. Subsequently, two samples were pretreated for 3 days and then leached using a nitrate-chloride-acid solution for 432 h. Figure 11a shows the effect of pretreatment on copper extraction when stirred leaching was carried out. After 432 h, the copper extractions achieved were 26.8, 55.9 and 64.7% when comparing the test without pretreatment, with pretreatment of Test 13 and pretreatment of Test 14, respectively. The results show the positive effect that the use of a pretreatment stage on the chalcopyrite dissolution, where after 96 h the extraction of copper without pretreatment was less than 10%, well below that obtained when the pretreatment was conducted, where 38 and 45% Cu extractions were obtained. This increase is due to the fact that the chalcopyrite leaching is accelerated with the pretreatment due to the high ionic load present in contact with the mineral as noted in other studies [13,15,34]. Therefore, after 96 h the copper

extraction with pretreatment (45.1%) exceeds that after 432 h without pretreatment (26.8%). The ORP values during the tests were in a range between 765 mV and 808 mV. No effect of pretreatment on redox potential values was observed. The pH values were low 0.4, during the leaching tests. Figure 11b shows molar ratios where the values are close to 1:1 in the tests carried out.

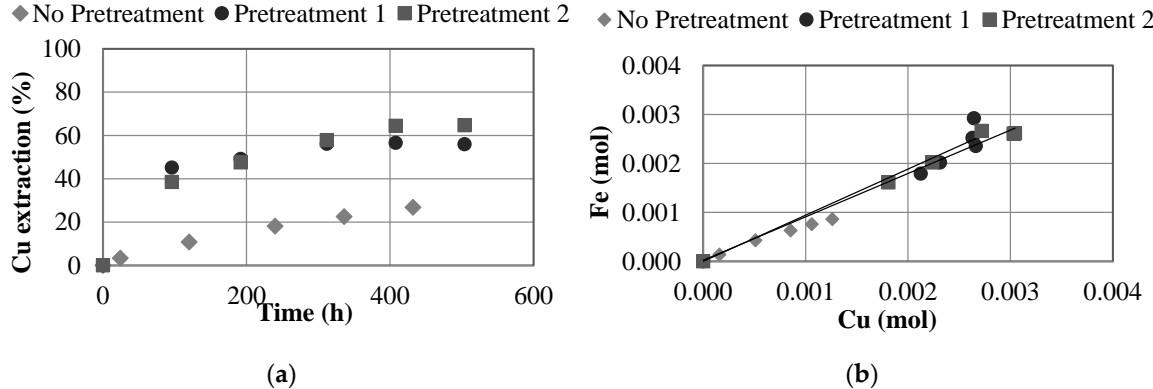

(a) (b)

**Figure 11.** (**a**) Copper extraction (%) versus time (h) with pretreatment conditions. (**b**) Cu/Fe molar ratio with pretreatment conditions. Experimental conditions: [NaNO$_3$] = 0.7 M, [H$_2$SO$_4$] = 0.7 M, seawater as dissolvent, −150 + 75 μm, S/L ratio 2 g/L, 45 °C, 350 rpm, sample A.

### 3.2. Leaching Kinetic

The kinetic model (Equations (2) and (3)) used was proposed by Sokić et al. [21]. This model represents a mixed control (diffusion and surface reaction). Where: α = converted fraction, t = time (h), τ = time at total conversion (h), T = temperature (K), k= kinetic coefficient obtained by slope of model (h$^{-1}$).

$$\frac{t}{\tau} = -Ln\,(1-\alpha) \tag{2}$$

$$\alpha = \frac{\%\ copper\ extraction}{100} \tag{3}$$

Figure 12 shows plots of the kinetic model with time at different temperatures and the Arrhenius plot using the copper extractions obtained at different temperatures.

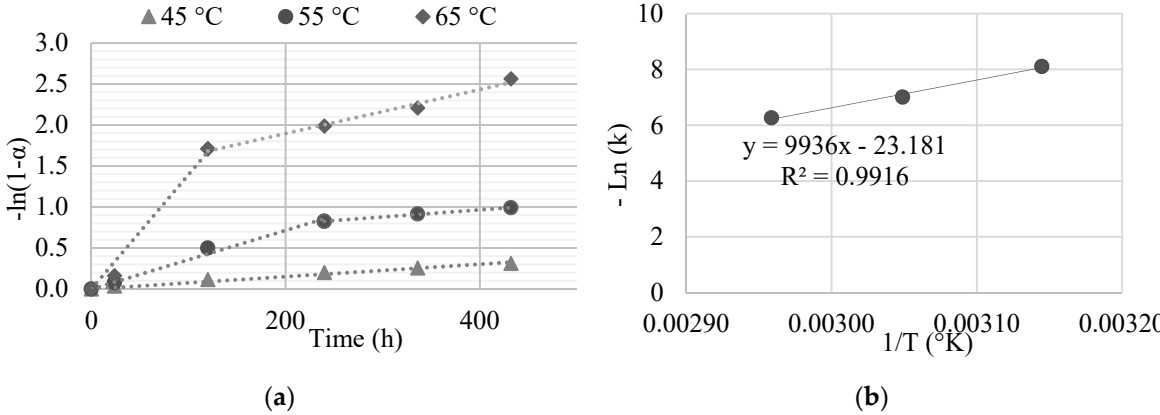

(a) (b)

**Figure 12.** (**a**) Kinetic model versus time (h) at different temperatures. (**b**) Arrhenius plot.

In the initial times, the dissolution of chalcopyrite could be limited by chemical reaction, due to low particle size. With the increase of time, the diffusion of reagents from solution to solid is limited by the passivate layer. Using the Arrhenius equation, the calculated activation energy was of 82.6 kJ/mol (Figure 12b, the Arrhenius plot correspond to when surface reaction control of the mechanism is dominant). The high value obtained for the activation energy indicates kinetic control by chemical

reaction that is sensitive to temperature. Sokić et al. [21] determined an activation energy of 83 kJ/mol for chalcopyrite leaching in nitrate-acid medium, a value similar to that obtained in this study.

## 4. Conclusions

Chalcopyrite leaching was studied in a nitrate-chloride acid media at different conditions. The results showed:

- The presence of sodium nitrate in chalcopyrite leaching using seawater produces higher recoveries of copper and iron with an ideal concentration between 0.25 and 0.5 M when 0.7 M of sulfuric acid was used at 45 °C.
- Increasing the concentration of sulfuric acid did not show a significant effect on chalcopyrite leaching using seawater between 0.25 and 0.7 M. A concentration between 0.25 and 0.5 M should be enough to provide acid and oxidative media for copper dissolution when 0.7 M of sodium nitrate was used at 45 °C.
- The solid-liquid ratio and temperature variables both showed positive effects when increased using equivalent experimental conditions. The decrease in particle size produced a greater contact surface allowing greater dissolutions of copper and iron.
- The increase of chloride concentration from 0 to 60 g/L, produced an increase in the dissolution of Cu and Fe, reaching 64% Cu at 60 g/L Cl⁻ when 0.7 M sodium nitrate and 0.7 M sulfuric acid were used at 45 °C.
- The use of seawater and waste brine were beneficial for chalcopyrite leaching when compared to deionized water.
- Leaching of different chalcopyrite samples showed two different dissolution behaviors, 27 and 64% copper extractions. This could be attributed to the particle size distribution and degree of release of the mineral in the particle.
- The use of a pretreatment had positive effects in terms of the processing time and the copper and iron extractions. This is due to the acceleration of the dissolution of the mineral during the curing time. Copper extraction of 64.7% was obtained when a pretreated mineral was leached in comparison to when pretreatment was not performed with only 26.8% Cu extraction obtained under the same leaching conditions.
- The kinetic model used showed that the chemical reaction controls the dissolution of copper and iron and this highly influenced by temperature. The activation energy obtained was 82.6 kJ/mol.
- SEM/EDS confirmed elemental sulfur formation. This is related to the formation of a layer of elemental sulfur in the chalcopyrite particles.

This study presented an alternative method of leaching chalcopyrite ores using seawater, discard brines and caliche waste (solid or solution) at 45 °C. The nitrate-chloride-acid system showed promising results at laboratory scale. A pretreatment stage before leaching should be included in a future process enhance leaching efficiency.

**Author Contributions:** Conceptualization, P.H. and G.G.; experimental design, P.H. and G.G.; investigation, G.G.; validation, N.T. and J.C.; formal analysis, P.H., G.G., N.T., J.C. and M.M.; writing—original draft preparation, P.H. and M.M. All authors have read and agreed to the published version of the manuscript.

**Funding:** This research was funded by ANID-Chile through Fondecyt de Iniciación Project N°11170179 and Project ING2030 CORFO Code 16ENI2–71940.

**Acknowledgments:** The authors thank to Laboratorio de Investigación de Procesos of Universidad de Antofagasta for the support provide.

**Conflicts of Interest:** The authors declare no conflict of interest.

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
