# Peer review of "Caliche and Seawater, Sources of Nitrate and Chloride Ions to Chalcopyrite Leaching in Acid Media"

_metals, doi:10.3390/met10040551_

Round 1
Reviewer 1 Report
The authors investigated the leaching of copper from chalcopyrite using nitrate-chloride-acid media using seawater or brine to provide chlorine to determine the effect of sulfuric acid, nitrate and chloride concentration, type of dissolvent, temperature, solid/liquid ratio, particle size, sample of mineral and sample pretreatment.
The manuscript is an interesting and the investigated issue can be of interest of the readers of Metals. In general, this is well structured and the experiments are well designed and cover a wide spectrum of variables. However some corrections are needed. I suggest the next minor changes:
Introduction
Line 38. I suggest to change “a mineral that is present in almost all copper deposits “ by “chalcopyrite (CuFeS2) is the most common copper ore”
Line 48. Please, specify if this datum is for Chile.
Line 66. These discarded salts can be used as oxidant in chalcopyrite
Samples and methods
Why did you select these 5 samples?
Mineralogy. How did you quantify the minerals?. The Siemens/Bruker, Semi-QXRD, model D5000 is the diffractometer, but, which where the analytical conditions and how did you quantifly the diffractograms?.
Line 101. You cannot write this “Mineralogical species of chalcopyrite samples”, you should say something like “Mineralogical composition of the studied samples” or “Mineralogy of the chalcopyrite ores used…”.
In Table 2, the fist column is chemical composition, you could include and additional column with the mineral names (Chalcopyrite, Pyrite, ?, Albite, Anhydrite, Gypsum, Muscovite)
K0.4Na0.6Cl ??. K and Na size of atoms is very different and both elements do not are together in chloride structures. I think that probably here there are halite (NaCl) and sylvite (KCl)
Results and discussion
I find this section correct but some improvements could be introduced.
Line 250 Indicate the experiment number of image in Figure 8.
Images of Figure 8 could be improved. This should be easier to read if you label them as A, B, C and D. The lower spectra belong to the whole corresponding images or just one point or a window’. Please indicate this. I understand that it is a point, since there is almost no Cu left and practically no Fe
Line 264. Instead of “Effect of chalcopyrite sample” another title without the Word simple should be more appropriate, e.g. “Effect of chalcopyrite grade “
Line 266. “grouped into two groups”, better: “classified into two groups”.
Figure caption of Figure 10. …. (c) Particle size distribution of samples: A, B, C, D and E after the experiments.
Line 272. “The differences observed in both groups can be attributed to the degree of chalcopyrite release in the mineral,” chalcopyrite is a mineral, then, do not use mineral in this sentence but …in the rock.
To do enrich this discussion it should be interesting if you present a brief description of the textures in each simple in the samples subsection, probably this could be illustrated with some optical microscope images.
The section of conclusions is correct.
References
The number of references is correct but:
- 1. They do not are listed following the instructions of MDPI journals: e.g. for journal articles: Author 1, A.B.; Author 2, C.D. Title of the article. Abbreviated Journal Name Year, Volume, page range.
-some mistakes are in the list, e.g. in reference 2, 19-27,30-37 and more all authors must be listed.

Author Response
Dear Reviewer:
Thanks for your revision. All suggestions and corrections were done according to pdf document and your comments. We list the changes made:
- Changes in lines 37-38, 48, 66, 101, 250, 264, 266, 272 were done.
- Table 2 was corrected.
- Figure 8 was corrected and the caption figure was improved.
- Figure 10 was corrected.
About the questions, we answered as follows:
Reviewer: Why did you select these 5 samples?
Answer: The five samples were obtained from different copper mines from Atacama and Antofagasta region of Chile. These samples were chosen due the homogeneus distribution in a map, so the samples could be representatives from different zones of northern of Chile.
Reviewer: Mineralogy. How did you quantify the minerals?. The Siemens/Bruker, Semi-QXRD, model D5000 is the diffractometer, but, which where the analytical conditions and how did you quantifly the diffractograms?.
Answer: The model of equipment was corrected. This equipment provide a semi cuantitative result using TOPAS (total pattern anlysis software) to cuantify the sample.
Reviewer: To do enrich this discussion it should be interesting if you present a brief description of the textures in each simple in the samples subsection, probably this could be illustrated with some optical microscope images.
Answer: The suggestion made is totally consistent. Using QEMSCAN the degree of release of the particle could be clearly seen. Unfortunately, this analysis was not performed, and was not included in the work. Therefore, we cannot add it at this time.
We appreciate the review and look forward to any further improvements that may need to be made.
Best regards
Reviewer 2 Report
This is a very interesting paper and addresses an area that I have been curious about. There are just a few items that I think should be addressed to make it a stronger paper.
- It is not clear whether any replicates were conducted to make sure that the results are reproducible. While the trends in general appear pretty strong, it would be good to select a few of the experiments to repeat exactly to confirm that the same results are achieved in the replication.
- On page 6, figure 2, the graph shows a higher percent copper dissolution as the quantity of solids per liter is increased, which is the opposite of what is normally seen. The authors note as a possibility that this might have been due to larger amounts of ferric iron in solution at the higher solids concentration, which would lead to faster copper dissolution. Could this be checked by repeating the 2g/liter experiment, dosed with supplemental ferric chloride to give it the amount of ferric iron that was present in the 10g/liter experiment, to see how much that affects the dissolution rate? If ferric iron is beneficial, it would be a very useful addition to the leaching solution.
Author Response
Dear Reviewer:
Thanks for your revision and suggestions. About the suggestions:
Reviewer: It is not clear whether any replicates were conducted to make sure that the results are reproducible. While the trends in general appear pretty strong, it would be good to select a few of the experiments to repeat exactly to confirm that the same results are achieved in the replication.
Answer: Only three tests were performed in duplicate: tests 5, 6 and 10. These tests were chosen at random to check the reproducibility of the tests. Since the duplicates agreed on their results, it was decided not to work with duplicates for the following tests. The results showed in tests 5, 6 and 10 correspond to average results with a standar deviation of ±1.086.
Reviewer: On page 6, figure 2, the graph shows a higher percent copper dissolution as the quantity of solids per liter is increased, which is the opposite of what is normally seen. The authors note as a possibility that this might have been due to larger amounts of ferric iron in solution at the higher solids concentration, which would lead to faster copper dissolution. Could this be checked by repeating the 2g/liter experiment, dosed with supplemental ferric chloride to give it the amount of ferric iron that was present in the 10g/liter experiment, to see how much that affects the dissolution rate? If ferric iron is beneficial, it would be a very useful addition to the leaching solution.
Answer: The suggestion made is totally consistent. We did not think about this propose. A new tests could be included according to suggestion of the reviewer. This work is parte of a project so, new tests are carring out. We considering this propose to a future tests.
We appreciate the review and look forward to any further improvements that may need to be made.
Best regards
Reviewer 3 Report
Your arguments for undertaking this work are sound and I am comfortable that the leaching results presented are reliable. Various aspects of the work could have been considered in greater detail, particularly if you are seeking to understand what the limitations of your approach are in terms of copper extraction from chalcopyrite. Presumably this will be a topic for future work, i.e. optimising the leaching of chalcopyrite.
There are parts of the manuscript where more information could be given or the discussion of specific aspects improved. In relation to the kinetic modelling the data seem to indicate that the model chosen is not the best available and I would encourage you to revisit that topic.
I have spent time addressing the English expression for the reason that this can be a distraction from what you are trying to say.
The attachment provided contains a range of suggestions and many queries for you to address which I believe will improve the quality of your manuscript.

Author Response
We upload an attachment.

Round 2
Reviewer 3 Report
Thank you for attending to my queries. I do have further comments/queries for you but these should be relatively straightforward to address. Some of the comments are simply my thoughts and are included for you to consider for future work. I agree that the model of Sokic et al is okay but please check the presentation of you data in Fig 12(a) to ensure it is consistent with their approach and confirm that the slopes used to estimate the activation energy are correct. Another matter I did not pick up previously is that your plots of mole Cu versus mole Fe are not always that close to 1:1 and there will be various reasons for this. You may wish to make comments on the deviations and perhaps suggest/hypothesize why the deviations occur.
My comment about the number of decimal places used can be rewritten as "if the error in the method is greater than the value of the last decimal place, you can't justify including that last decimal place". As a general rule analytical data can only be quoted to 3 significant figures. I would suggest talking to an analytical chemist if you need further clarification.
Best wishes with your future studies.

Author Response
We attached a file.
